# Capsaicin Ameliorates the Cyclophosphamide-Induced Cardiotoxicity by Inhibiting Free Radicals Generation, Inflammatory Cytokines, and Apoptotic Pathway in Rats

**DOI:** 10.3390/life13030786

**Published:** 2023-03-14

**Authors:** Rayan A. Ahmed, Mohammad Firoz Alam, Saeed Alshahrani, Abdulmajeed M. Jali, Abdullah M. Qahl, Mohammad Khalid, Hisham M. A. Muzafar, Hussain N. Alhamami, Tarique Anwer

**Affiliations:** 1Department of Pharmacology and Toxicology, College of Pharmacy, Jazan University, Jazan 45142, Saudi Arabia; 2Department of Pharmacognosy, College of Pharmacy, Prince Sattam Bin Abdulaziz University, Alkharj 16278, Saudi Arabia; 3Department of Pharmacology and Toxicology, College of Pharmacy, King Saud University, Riyadh 11451, Saudi Arabia

**Keywords:** cyclophosphamide, capsaicin, free radicals, oxidative stress, inflammatory cytokines, apoptosis

## Abstract

Cyclophosphamide is an antineoplastic agent that has a broad range of therapeutic applications; however, it has numerous side effects, including cardiotoxicity. Furthermore, chili peppers contain a substance called capsaicin, having antioxidant and anti-inflammatory effects. Thus, this research paper focuses on the potential mechanism of capsaicin’s cardioprotective activity against cyclophosphamide-induced cardiotoxicity by measuring the expression of oxidative and inflammatory marker such as interleukins and caspases. The following groups of rats were randomly assigned: only vehicle given for 6 days (control group); cyclophosphamide 200 mg/kg intraperitoneal on 4th day only (positive control group); capsaicin 10 mg/kg orally given for 6 days followed by cyclophosphamide 200 mg/kg on 4th day of treatment; capsaicin 20 mg/kg orally for six days followed by cyclophosphamide 200 mg/kg on 4th day of treatment; and maximum amount of capsaicin alone (20 mg/kg) orally for six days. Using ELISA kits, it was found that the cyclophosphamide administration significantly increased the levels of lactate dehydrogenase, troponin-I (cardiac cell damage marker), lipid peroxidation, triglyceride, interleukin-6, tumor necrosis factor-alpha, and caspase 3. However, it markedly reduced the antioxidant enzymes catalase and glutathione levels. Both doses of capsaicin could reverse cardiac cell damage markers, as shown by a significant decline in (lactate dehydrogenase and troponin-I). In addition, capsaicin significantly reduced the cytokine levels (interleukin-6 and tumor necrosis factor-alpha), caspase 3, lipid peroxidation, and triglycerides. However, capsaicin treatment significantly raised the antioxidant content of enzymes such as glutathione and catalase. The capsaicin-treated group restored the oxidative parameter’s imbalance and generated considerable protection against cardiomyocyte harm from cyclophosphamide in male Wistar rats. These protective effects might be beneficial against the negative impacts of cyclophosphamide when used to treat cancer and immune-mediated diseases.

## 1. Introduction

Anticancer drugs associated with cardiotoxicity seriously threaten the health of cancer patients. Therefore, cardio-oncology is becoming a severe health problem in the contemporary period [1]. The most well-known chemotherapeutic drugs that induce cardiotoxicity are anthracyclines antibiotics, alkylating agents such as cisplatin, cyclophosphamide (CYC), ifosfamide, carmustine, chlormethine, busulfan, and mitomycin [2]. Therefore, predicting cardiotoxicity helps us intervene early and provide effective treatments to reduce morbidity and fatality rates.

Cyclophosphamide (CYC) is a nitrogen mustard replaced with oxazaphosphorine that has strong cytotoxic and immunosuppressive properties [3,4]. Cyclophosphamide is used in cancer and other diseases, such as rheumatoid arthritis, Burkitt’s lymphoma, systemic lupus erythematosus (SLE), multiple sclerosis (MS), neuroblastoma, leukemia, multiple myeloma, endometrial cancer, breast cancer, lung cancer, and Hodgkin’s and non-Hodgkin’s lymphoma. According to the CYC mode of action, an alkyl moiety is covalently attached when the drug’s electrophilic nature interacts with the nucleophilic nature of proteins or DNA molecules. Moreover, CYC as a prodrug must be first be metabolized by liver cytochromes (P450) into phosphoramide mustard and acrolein, producing its actions. Whereas acrolein induces the cytotoxic effects of CYC (hemorrhagic cystitis) through the generation of free radicals, phosphoramide mustard is in charge of crucial anticancer activity and (DNA alkylating agent) [5]. Individuals receiving (>150 mg/kg or 1.55 g/m^2^/day) are highly susceptible to cardiotoxicity [6]. However, cumulative doses are the key contributor to the induction of other toxicities, including gonadal toxicity, bone marrow suppression, and cancer [7]. Although there has been no CYC toxicity at doses (<100 mg/kg), the lowest dose to induce cardiotoxicity is not yet known [8]. The hematologic and pathologic consequences of several high-dose CYC regimens on rhesus monkeys were examined by Storb and colleagues (1970). More than 75% of the monkeys administered (240 mg/kg for 1–4 days) of CYC experienced cardiotoxicity. Furthermore, Santos et al. reported the first instance of fatal CYC-induced cardiotoxicity in a human (1971). Since 1971, several studies have revealed that individuals who receive (120–270 mg/kg for 1–8 days) of CYC may experience cardiotoxicity [9,10]. It is reported that CYC toxic effect is irreversible and may cause hemorrhagic cell death when administered in a (single/large dose), leading to heart failure [11]. Cardiotoxicity caused by CYC might appear as early as 48 h after taking medicine but can last up to 10 days [8]. Circulatory cardiac markers may allow us to predict chemotherapy-induced early heart cell damage. For example, B-type natriuretic peptide (BNP) may be the most effective measure in the context of cyclophosphamide-induced acute heart failure since it is elevated on day one and remains elevated for seven days of treatment [12,13,14]. In addition, troponin I or T are very sensitive and precise plasma indicators of cardiomyocyte damage that may also be beneficial in monitoring cyclophosphamide-induced cardiotoxicity. After the administration of high-dose CYC, the troponin level typically peaks 8 to 15 days later [15]. The mechanisms underlying cyclophosphamide-induced cardiotoxicity are not yet well-known.

Various studies show that medical plants demonstrated cardioprotective efficacy against cardiotoxicity produced by cyclophosphamide (summarized by Ayza et al.) [16]. For example, the antioxidant activity of spices and herbs is attributed to the existence of antioxidants. such as flavonoids, flavones, isoflavones, catechins, isocatechins, and anthocyanin compounds. Additionally, it was revealed that the vasodilator drugs (nicorandil) and the xanthin oxidase inhibitors (allopurinol and febuxostat) reversed the heart damage induced by CYC in male Wistar rats (summarized by Ayza et al.) [16].

Humanity may have been aware of capsaicin-containing foods as early as (7000–9000 BP) [17,18]. Capsaicin (Caps) is a herbal compound (alkaloid) derived from chili peppers. It is used as a flavoring agent (spice), coloring agent [19], and a topical medication worldwide. The main pungent phytochemical, “capsaicin”, is discovered in the fruits (genus Capsicum and family Solanaceae) [20,21]. The chemical formula for capsaicin (Caps) is C_18_H_27_NO_3_. It is a crystalline lipophilic phytochemical that has no flavor or odor. Caps is soluble in fat, alcohol, and oils and has a molecular weight (305.40 g/mol). Three different Hydroxy-capsaicin derivatives have been defined as the primary metabolites produced upon the metabolism of Caps. In terms of mode of action, Caps activates the transient receptor potential vanilloid 1 (TRPV1) channel, mainly found in the sensory neurons [22,23,24]. The mature pepper fruit is high in phenolic compounds, especially flavonoids, which have antioxidants and other bioactive substances. It is also high in carotenoids, which have antioxidant effects [25,26]. Caps’ powerful pharmacological effects include analgesic activity, anti-obesity, antipruritic, anti-inflammatory, anti-apoptotic, antineoplastic, free radical scavenging, cardioprotective, hepatoprotective, and neuroprotective actions [20,21,27,28]. In terms of molecular mechanisms, a wide range of signaling pathways are regulated by Caps treatment, including pro-inflammatory chemicals, such as prostaglandin E2 (PGE2), (IL-6), tumor necrosis factor-alpha (TNFα), and nitric oxide (NO). Likewise, it was verified that Caps plays a role in lowering plasma levels of advanced oxidation protein products (AOPP), malondialdehyde (MDA), and reactive oxygen species (ROS) [27,29]. Eventually, the need for more studies of the pharmacological properties of Caps is influenced by the lack of essential medications for several disorders (treatment or prophylaxis) [21].

Therefore, this research concentrates on investigating the antioxidant and anti-inflammatory properties of Caps and possible molecular mechanisms in preventing or alleviating cyclophosphamide-induced cardiotoxicity. Furthermore, it will emphasize the cardioprotective properties of naturally occurring plant-derived Caps as an antioxidant, which has recently attracted much attention.

## 2. Materials and Methods

### 2.1. Chemicals and Reagents

The current investigations were carried out using materials of the highest analytical grade that were commercially available. Capsaicin and cyclophosphamide were bought from (Sigma Aldrich Co, Old Brickyard, New Rd, Gillingham, SP8 4XT, UK). Elisa kits for lactate dehydrogenase (LDH), lipid peroxidation (LPO) (troponin-I, TNFα, IL-6, TG, and caspase 3) were bought from (MyBioSource, Inc. Post Box 153308, San Diego CA 92195-3308, USA.). Total proteins and serum cardiac biomarkers were assessed using a kit obtained from (Randox Laboratories Ltd., 44Largy Rd, Crumlin BT29 4RN, UK).

### 2.2. Animal Model and Study Design

Male Wistar rats weighing (150–220 g) were obtained from the Medical Research Center (MRC) at Jazan University, Saudi Arabia. Following standard laboratory procedures (American Association for Laboratory Animal Science), rats were housed in animal housing (College of Pharmacy, Jazan University, Jazan, Saudi Arabia). Before conducting the studies, animals were housed for seven days in solid-bottom polycarbonate cages with rice husk used as bedding. Rats were given unlimited access to water and a regular laboratory diet (autoclaved).

Five test groups of six rats were created, as follows:

Control: Only vehicle given for six days (normal saline).

Positive control: CYC 200 mg/kg, intraperitoneal (i.p) on the 4th day with normal saline [28,30,31] (CYC 200 alone).

Treatment 1: Caps 10 mg/kg by mouth (p.o) for six days and a single dose of CYC 200 mg/kg, i.p on the 4th day of treatment [32,33] (Caps 10 + CYC 200).

Treatment 2: Caps 20 mg/kg, p.o; for six days and a single dose of CYC 200 mg/kg, i.p on the 4th day of treatment [32,33] (Caps 20 + CYC 200).

Treatment 3: The maximum dosage of Caps (20 mg/kg. p.o. for six days) (Caps 20 only).

The rats underwent a minimum of 12 h fasting on day eight. After that, all animals were subjected to chloral hydrate to make them all unconscious. To perform cardiac function tests, blood was promptly withdrawn from (the retro-orbital plexus) of each rat in each group and centrifuged at (3000 rpm for 15 min). The serum was then gathered and kept at −20 °C for later assessments of (LDH, troponin-I, and TG). After that, all animals were sacrificed, and the hearts were taken out. First, heart tissues were homogenized using a 10% child PB solution (0.1 M and pH 7.4). Then, the homogenate was centrifuged (3000 rpm for 15 min at 4 °C). The upper portion was used for the cytokine assay and estimation of LPO, while the remaining antioxidants catalase (GSH, CAT) were determined in post-mitochondrial supernatant (PMS) using the 10% homogenate separated at 10,500 g for 30 min at 4 °C.

### 2.3. Assessment of Serum Cardiac Function Level (LDH)

The LDH test was performed by MyBioSource’s standard procedure utilizing the double antibody sandwich ELISA method (LDH; cat. no. MBS2018912; MyBioSource, Inc.). The pre-coated antibody was a (monoclonal) anti-rat LDH antibody, whereas the detection antibody was a (biotinylated polyclonal) anti-rat LDH antibody. The wells of an ELISA plate were washed with PBS or TBS after adding samples and biotinylated antibodies to the appropriate wells. After that, Avidin-peroxidase conjugates were then introduced to each well. After washing with PBS or TBS, the TMB substrate changes the color of the well after the enzyme conjugate has been completely removed. The TMB then generates a (blue product) from the peroxidase activity, which becomes (yellow) once the stop solution is added (Reagent C). The color intensity was measured at (450 nm).

### 2.4. Assessment of Serum Cardiac Function Level (Troponin-I)

An established protocol by (MyBioSource, USA) was used to quantitatively assess the amount of (cTnI) (cTnI; cat. no. MBS7224954; MyBioSource, Inc.). The single-step sandwich test employs two monoclonal antibodies against (cTnI) (measuring range 0.03 to 50 ng/mL) and paramagnetic beads coated (solid phase). In addition, a second-generation cardio-specific test (produced by Elecsys^®^, Troponin T STAT immunoassay) was used to assess (cTnT).

### 2.5. Assessment of Cardiac Tissue Lipid Peroxidation Level (LPO)

Malondialdehyde (MDA), an indicator of LPO, was used as the basis for the LPO assay, with some modifications from the Utley et al. method [34,35]. Thiobarbituric acid (0.67%) and 10% trichloroacetic acid made up the mixture. The supernatant was produced by centrifugation at 3000× *g* for 15 min at 4 °C. Samples were then held in 100 °C water for (10 min) after the supernatant was separated. The reading of the samples was taken at 535 nm (after cooling). The amount of LPO was calculated as nmoles of MDA/g tissue using the molar extinction coefficient (MEC) of 1.56 105 M^−1^ cm^−1^.

### 2.6. Assessment of Cardiac Tissue Glutathione Level (GSH)

Glutathione (GSH) was evaluated according to the Jallow et al. technique [36]. First, sample PMS was precipitated using 4% sulfosalicylic acid (1:1 ratio). Then the mixture was centrifuged (3000 rpm) after being incubated (1 h/4 °C) to separate the supernatant. Next, sodium phosphate buffer (Na PB 2.1 mL of 0.1 M) at pH (7.4), with supernatant (0.4 mL) and 0.4 mL of di-thibis-nitrobenzoic acid, were added to the 3 mL assay sample. The absorbance of the sample was immediately recorded (412 nm). Finally, the GSH concentration in mole/g tissue was determined using the molar extinction coefficient (MEC) (1.36 104 M^−1^ cm^−1^).

### 2.7. Assessment of Cardiac Tissue Catalase Activity (CAT)

The Claiborne et al. technique was used to quantify CAT activity [37]. First, phosphate buffer of PMS (50 μL) and 1.95 mL of 6.0 mM hydrogen peroxide (H_2_O_2_) make up the reaction volume. Next, the activity of CAT is measured in nmol of H_2_O_2_ consumed/min/mg/protein using a 43.6 103 M^−1^ cm^−1^ molar extinction coefficient (MEC) (MEC). Finally, the reaction absorbance was quantified at (240 nm).

### 2.8. Assessment of Serum Lipid Profile Level (TG)

As previously described, an ELISA kit for triglycerides (TGs) with an established protocol by (MyBioSource, USA) was used to quantitatively assess the amount of rat TG (lipid profile marker) (TG; cat. no. MBS726298; MyBioSource, Inc.). The detection antibody used was a (biotinylated polyclonal) anti-rat TG antibody, whereas the pre-coated antibody was a (polyclonal anti-TG antibody). After adding samples and biotinylated antibodies to corresponding wells of an ELISA plate, the reaction will occur, and the color will develop. Then the stop solution will be added to measure the developed color at (450 nm).

### 2.9. Estimation of Cardiac Tissue Pro-Inflammatory Cytokines (IL-6 and TNFα) and Caspase 3 Levels

ELISA kits were utilized per the manufacturer’s guidelines to ascertain the level of pro-inflammatory cytokines (IL-6 and TNFα), and a calorimetric estimation kit was used to assess caspase 3. For example, the absorbance of interleukin (IL-6 and TNFα) was recorded at (450 nm) using an ELISA microplate reader (BioTek ELx800, BioTek^®^ Instruments, Inc. Highland Park, P.O. Box 998 Winooski, VT, 05404-0998 USA), whereas caspase 3 absorbance was measured at (405 nm).

### 2.10. Statistical Analysis

The data were statistically analyzed using analysis of variance (ANOVA), followed by Tukey-test Kramer’s. The average standard error of six rats’ mean (SEM) was used to illustrate the results. The *p* < 0.05 was used to evaluate statistical significance data.

## 3. Results

### 3.1. The Cardioprotective Effects of Capsaicin via Regulation of Lactate Dehydrogenase Level (LDH)

After receiving 200 mg/kg of CYC, the level of LDH was significantly increased, reaching approximately 1000 U/L compared to the control group (less than 250 U/L). In the presence of CYC, pre-administration of (10 mg/kg) Caps slightly decreased the content of (LDH) as to the cyclophosphamide-treated group. However, the administration of (20 mg/kg) Caps substantially decreased the (LDH) to a point close to the control group. Furthermore, in the Caps only treated group (20 mg/kg), the content of (LDH) was not significantly changed in comparison to the control (Figure 1).

### 3.2. The Cardioprotective Effects of Capsaicin through Regulation of Troponin-I Concentration

After receiving 200 mg/kg of CYC, the concentration of troponin-I in the serum was significantly elevated to approximately 50 (pg/mL) as to the control (<20 pg/mL). In the presence of CYC, the dose-dependent pre-administration of Caps (10 and 20 mg/kg) significantly decreased the level of (troponin-I) as to the positive control. Furthermore, in the Caps only treated group (20 mg/kg), the content of (troponin-I) was not significantly changed in comparison to the control (Figure 2). Furthermore, in the Caps treated group (20 mg/kg) only, the content of (troponin-I) was significantly decreased as compared with the group. Nevertheless, the difference in troponin-I levels between the Caps only treated group (20 mg/kg), and the control group was insignificant (Figure 2).

### 3.3. The Cardioprotective Effects of Capsaicin through Regulation of Lipid Peroxidation (LPO)

After receiving 200 mg/kg of CYC, the level of MDA indicating the lipid peroxidation (LPO) in the cardiac tissue was significantly increased, double the amount of the control group. In the presence of CYC, pretreatment with Caps induced a notable dose-dependent decrease in the level of (MDA) as to the cyclophosphamide-treated group. Furthermore, in the Caps treated group (20 mg/kg) only, the content of (MDA) was not significantly decreased as compared with the group (Figure 3).

### 3.4. The Cardioprotective Effects of Capsaicin through Regulation of Glutathione (GSH) Level

After receiving 200 mg/kg of CYC, the glutathione (GSH) enzyme level was significantly decreased by approximately 50% as to the control. In the presence of CYC, the dose-dependent pre-administration of Caps (10 and 20 mg/kg) significantly elevated the depleted level of (GSH) as to the cyclophosphamide-treated group. Furthermore, in the Caps treated group (20 mg/kg) only, the content of (GSH) was not significantly decreased as compared with the group (Figure 4).

### 3.5. The Cardioprotective Effects of Capsaicin through Regulation of Catalase (CAT) Level

After receiving 200 mg/kg of CYC, the catalase (CAT) enzyme level significantly decreased from around 30 to 10 nmol of H_2_O_2_/min/mg/protein. In the presence of CYC, the dose-dependent pre-administration of Caps (10 and 20 mg/kg) significantly increased the depleted level of (CAT) as to the group that received cyclophosphamide alone. Furthermore, in the Caps treated group (20 mg/kg) only, the content of (CAT) was not significantly changed as compared with the group (Figure 5).

### 3.6. The Cardioprotective Effects of Capsaicin through Regulation of Triglycerides (TG) Level

After receiving 200 mg/kg of CYC, the level of triglycerides (TG) was significantly increased with > 2 mg/dL in comparison to the control group (around 1.5 mg/dL). In the presence of CYC, the dose-dependent pre-administration of Caps (10 and 20 mg/kg) significantly decreased the level of TG compared to the group that received cyclophosphamide alone. Furthermore, in the Caps treated group (20 mg/kg) only, the content of (TG) was not significantly changed compared with the group (Figure 6).

### 3.7. The Cardioprotective Effects of Capsaicin through Regulation of Inflammatory Markers Production (IL-6, TNFα)

After receiving 200 mg/kg of CYC, the levels of inflammatory markers were significantly increased (IL-6: 3 folds) and (TNFα: 5 folds) as to their controls. Though in the presence of CYC, the dose-dependent pre-administration of Caps (10 and 20 mg/kg) significantly decreased these inflammatory markers (IL-6, TNFα) compared to the respective positive control groups. Furthermore, in the Caps treated group (20 mg/kg) only, the content of inflammatory marker (IL-6) was significantly changed (*** *p* = 0.001) compared with the control group (Figure 7a). Nonetheless, the amount of TNFα was not (Figure 7b).

### 3.8. The Cardioprotective Effects of Capsaicin through Regulation of Caspase 3 Level

After receiving 200 mg/kg of CYC, the concentration of caspase 3 was significantly elevated (quadrupled) as to the control group. In the presence of CYC, the dose-dependent pre-administration of Caps (10 and 20 mg/kg) significantly decreased the level of caspase 3 as to the group that received cyclophosphamide alone. Furthermore, in the Caps treated group (20 mg/kg) only, the content of caspase 3 was not significantly changed compared with the group (Figure 8).

## 4. Discussion

The nitrogen mustard alkylating agent, such as CYC, has strong anticancer, immunosuppressive, and immunomodulatory effects. Cardiomyopathy caused by chemotherapy is a vital issue in patient morbidity and mortality. A fatal side effect of using CYC, in particular, is heart cell damage [38,39]. Given anticancer treatments, such as CYC, the risk of cardiotoxicity increases in patients already suffering from other heart issues [38]. The cellular and biochemical changes associated with the cardiotoxicity of the CYC are attributed to the disturbance of lipid metabolism, production of free radicals, inflammation, and initiation of apoptosis. The results from this study revealed that a single dosage of CYC administration induced cardiac cell damage as indicated by an increase in the serum non-specific LDH enzyme and troponin-I level (a unique marker for the heart cell undergoing damage) [40]. Additionally, CYC increases lipid peroxidation, as shown by an alteration in malondialdehyde (MDA) level, and decreases antioxidants enzymes levels, both of which contribute to an abnormal lipid profile [41]. According to previous studies, exposing rats to CYC leads to kidney impairment and increases lipid peroxidation [42,43]. Furthermore, ROS levels induce inflammation, as shown by the increase in inflammatory mediators such as (ILs). However, in our current study, capsaicin (Caps) corrected these abnormal biochemical alterations induced by cyclophosphamide and did not have any effects when given alone. These results mean that Caps induced its effect in the presence of CYC.

In this study, we noticed that Caps reduces cyclophosphamide-induced cardiotoxicity by controlling oxidative stress, inflammation, and apoptosis. This might be the mechanism by which Caps exhibits its cardioprotective effects. First, a hazardous dose of CYC was given to the animals, which led to a substantial increase in the levels of cardiomyocyte death indicators such as LDH (over 1000 U/L) and troponin-I (50 pg/mL), compared to the respective control groups. Capsaicin therapy, however, was able to reduce the same markers in a dose-dependent manner. Then, we studied whether this effect was linked to a reduction in lipid peroxidation and ROS generation. In this project, we showed that CYC administration increased lipid peroxidation, as shown by an elevation in (MDA) level compared to the control group. An earlier finding revealed that carfilzomib produces oxygen-free radicals associated with increased MDA levels [44,45]. Oxidative stress occurs when ROS production rises to a level that outpaces cellular antioxidant capability [46]. Therefore, in this study, we measured the antioxidants enzyme levels (GSH and CAT) and found that CYC treatment considerably increased their levels compared to the control groups. However, Caps pretreatment lessened cyclophosphamide’s effects on these enzymes’ activity. These data show that the incline in the ROS generation could be associated with a disturbance in antioxidant levels. Moreover, the antioxidant defense system can be dysregulated by altering the plasma lipid profile [41]. Triglyceride (TG) level can be used as a parameter to check the lipid profile status. The findings in this study demonstrate that the TG level was substantially increased following the treatment with CYC and a dramatic dose-dependent decrease in the Caps pretreated group. These findings reveal Caps’ potent antioxidant ability that effectively mitigates the harm caused by the administration of CYC and its associated free radicals.

The interplay between oxidative stress and inflammation causes numerous diseases [47,48]. Inflammation can generate oxidative stress by activating many cell signaling pathways. For example, hydrogen peroxide (H_2_O_2_), a reactive oxygen species, can create the body’s inflammatory response by triggering nuclear factor kappa-B cell (NF-κB) [47,49]. Pro-inflammatory cytokines, such as interleukin (IL-6) and tumor necrosis factor-alpha (TNFα), produced by NF-κB expression, mediate immunological responses and inflammation [50]. IL-6 is a crucial cytokine for both pathogenic cardiac transition and cardiac protection. The IL-6 induces a cardioprotective effect on the cardiac myocytes during the initial reaction. Still, if its level is high and produced for a long time, it will cause hypertrophy and reduce the contractility of heart cells [51,52,53]. In this study, IL-6 and TNFα raised by more than three and five times after CYC administration. Due to the potent anti-inflammatory properties of Caps, we showed how these alterations in the cytokine levels were significantly diminished (dose-dependent manner). As mentioned before, NF-κB promotes the production of these cytokines and has other roles, such as inducing or inhibiting apoptosis (pro-apoptotic or anti-apoptotic functions) [54,55]. The variation in the NF-κB action is attributed to the type of cell and the extent to of the inflammation [55]. As the levels of cell death biomarkers were high following CYC treatment, we decided to study if the damage occurs to the cardiomyocyte via the caspase 3 enzyme, an executioner caspase that works at the end of the intrinsic apoptotic mitochondrial pathway [56].

A healthy mitochondrial physiology is essential for cell survival and well-being, and its disruption is linked to the etiology of disorders, including cardiovascular disorders. The mitochondria not only supply the cell with energy, but they also serve as the main source of ROS. The role of the mitochondria in cell death and stress responses has recently come to light. Cells have developed unique defenses against stress and inflammatory conditions for efficient cell activity and hemostasis. However, a number of polyphenols, including resveratrol, quercetin, and rutin, have been shown to target mitochondrial biogenesis both in vivo and in vitro [57]. In this study, we were interested to see whether pretreatment with Caps plays a role in controlling the caspase 3 level which is an apoptotic mitochondrial protein. Compared to the control group, the results reveal that CYC dramatically raises the amount of caspase 3. However, animals pretreated with Caps showed significantly reduced caspase 3 expression level in the heart tissue. As a result of caps’ potent antioxidant properties, ROS generation is reduced. This effect is connected to the activation of caspase 3, which provides a potential molecular defense mechanism for Caps against cardiotoxicity caused by cyclophosphamide.

Even though our study provided an interesting finding about the role of Caps in reducing the cardiotoxicity induced by cyclophosphamide, more studies should be conducted to obtain a deeper understanding. For example, Caps roles on gene expression levels utilizing immune-mediated diseases and cancer models are essential to be carried out. Additionally, more research can be conducted to test for other lipid profile markers, such as high-density lipoprotein (HDL), low-density lipoprotein (LDL), and very low-density lipoprotein (VLDL). Furthermore, measurements of other apoptotic genes, such as Bax and Bcl2 genes [58], and caspase proteins (intrinsic and extrinsic caspases) can also be conducted to better understand the molecular mechanism of Caps. Moreover, the early heart damage of cardiomyocytes can be determined by measuring B-type natriuretic peptide (BNP) since it is elevated on the first day of CYC treatment and continues to be elevated (up to a week) instead of troponin, which peaks (8–15 days) after the toxicity occurs.

## 5. Conclusions

In conclusion, capsaicin (Caps) is a potential candidate for reducing oxidative damage and minimizing the release of the inflammatory cytokines associated with cyclophosphamide-induced cardiotoxicity in rats. As cyclophosphamide used to treat cancer and immune-mediated disorders, this study indicated that the protective effects of Caps may be beneficial against the side effects of cyclophosphamide.

## Figures and Tables

**Figure 1 life-13-00786-f001:**
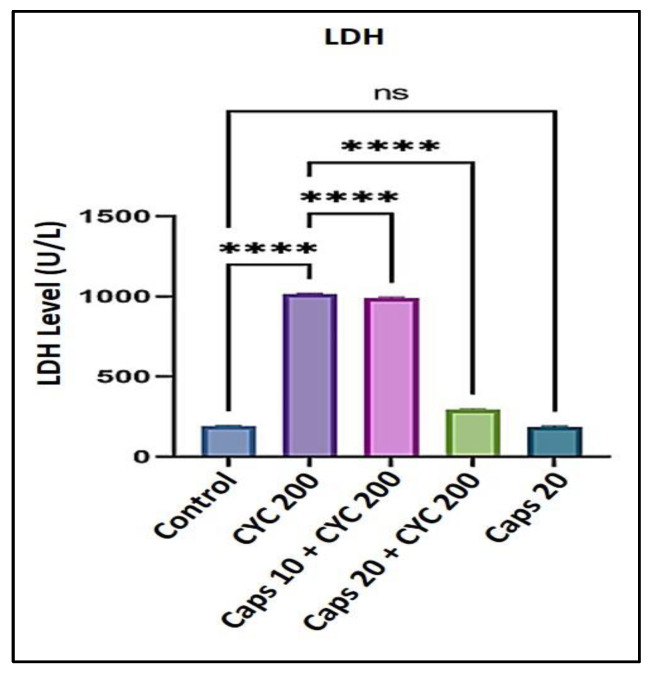
Protective effects of capsaicin on the dysregulated level of (LDH) against cyclophosphamide-induced cardiotoxicity. Measurement indicated as mean ± SEM (*n* = 6). A significant difference between the comparison groups is indicated by the symbol **** *p* = 0.0001. The non-significant difference between the groups was marked as (ns).

**Figure 2 life-13-00786-f002:**
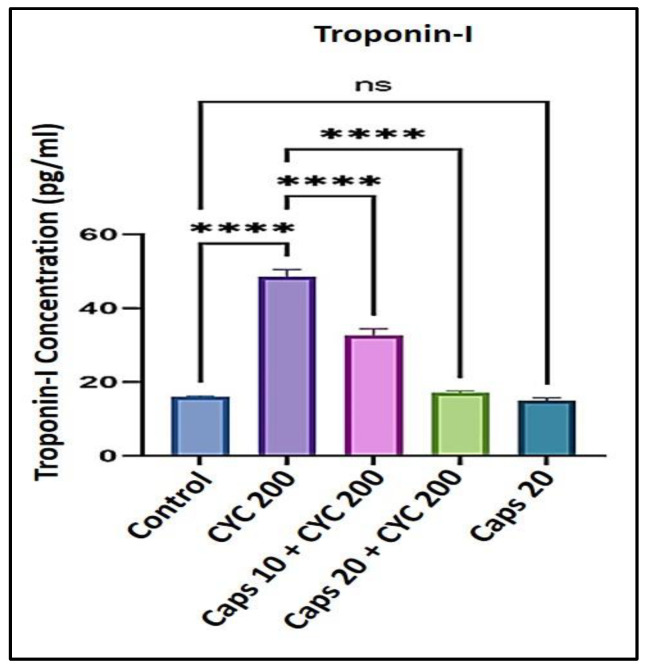
Protective effect of capsaicin on the dysregulated level of (troponin-I) against cyclophosphamide-induced cardiotoxicity. Measurement indicated as mean ± SEM (*n* = 6). A significant difference between the comparison groups is indicated by the symbol **** *p* = 0.0001. The non-significant difference between the groups was shown as (ns).

**Figure 3 life-13-00786-f003:**
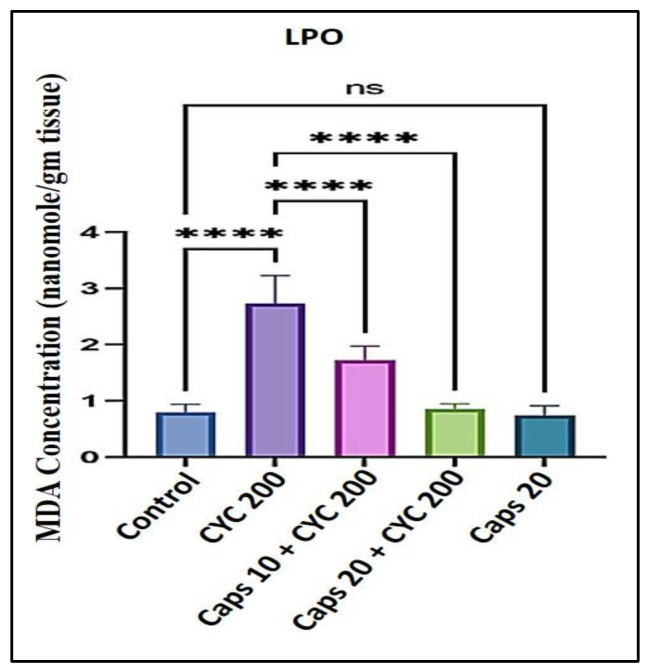
Protective effect of capsaicin on the dysregulated content of lipid peroxidation marker (MDA) against cyclophosphamide-induced cardiotoxicity. Measurement indicated as mean ± SEM (*n* = 6). A significant difference between the comparison groups is indicated by the symbol **** *p* = 0.0001. The non-significant difference between the groups was marked as (ns).

**Figure 4 life-13-00786-f004:**
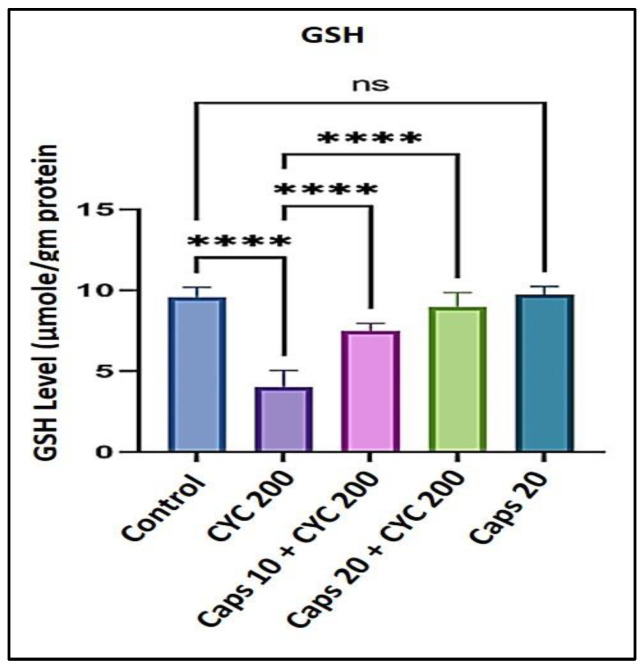
Protective effect of capsaicin on the dysregulated level of (GSH) against cyclophosphamide-induced cardiotoxicity. Measurement indicated as mean ± SEM (*n* = 6). A significant difference between the comparison groups is indicated by the symbol **** *p* = 0.0001. The non-significant difference between the groups was shown as (ns).

**Figure 5 life-13-00786-f005:**
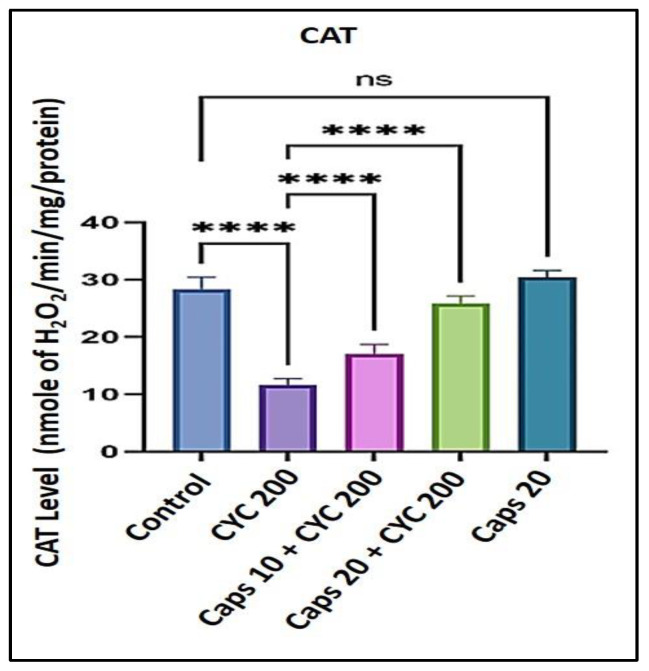
Protective effect of capsaicin on the dysregulated level of (CAT) against cyclophosphamide-induced cardiotoxicity. Measurement indicated as mean ± SEM (*n* = 6). A significant difference between the comparison groups is indicated by the symbol **** *p* = 0.0001. The non-significant difference between the groups was demonstrated as (ns).

**Figure 6 life-13-00786-f006:**
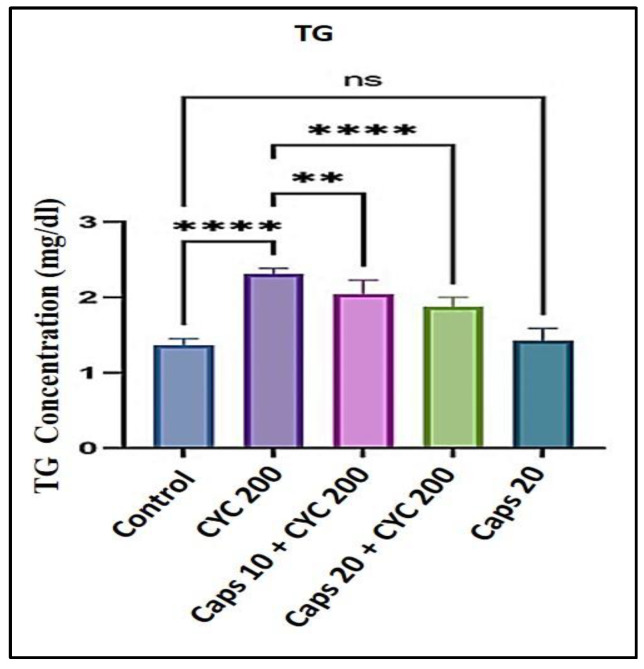
Protective effect of capsaicin on the dysregulated level of (TG) against cyclophosphamide-induced cardiotoxicity. Measurement indicated as mean ± SEM (*n* = 6). A significant difference between the comparison groups is indicated by the symbol **** *p* = 0.0001. ** *p* = 0.001. The non-significant difference between the groups was shown as (ns).

**Figure 7 life-13-00786-f007:**
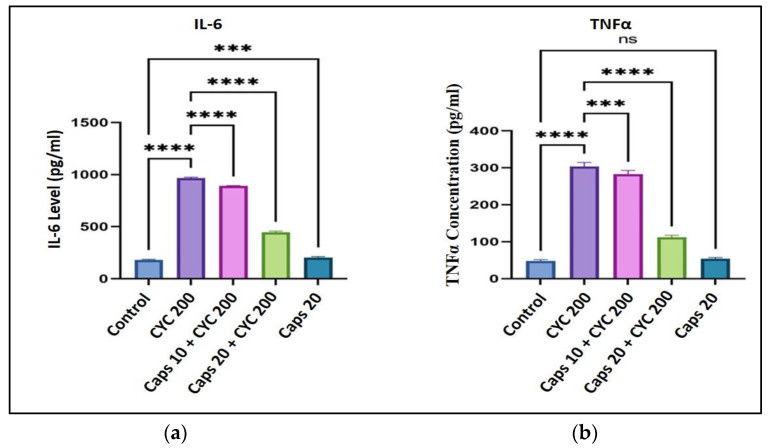
Protective effect of capsaicin on the dysregulated level of inflammatory markers IL-6 (**a**) and TNFα (**b**) against cyclophosphamide-induced cardiotoxicity. Measurement indicated as mean ± SEM (*n* = 6). A significant difference between the comparison groups is indicated by the symbol (**** *p* = 0.0001) and (*** *p* = 0.001). The non-significant difference between the groups was marked as (ns).

**Figure 8 life-13-00786-f008:**
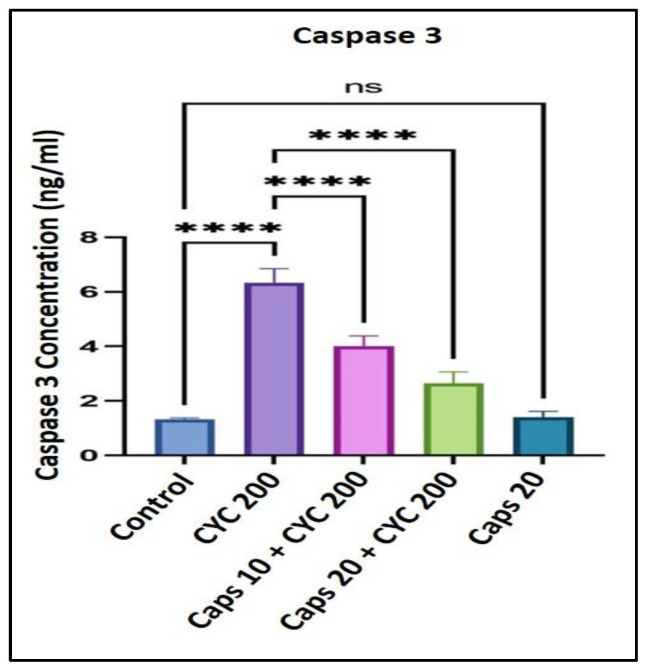
Protective effect of capsaicin on the dysregulated level of caspase 3 against cyclophosphamide-induced cardiotoxicity. Measurement indicated as mean ± SEM (*n* = 6). A significant difference between the comparison groups is indicated by the symbol **** *p* = 0.0001. The non-significant difference between the groups was marked as (ns).

## Data Availability

The authors confirm that the data are contained within the article.

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
