# Peer review of "Capsaicin Ameliorates the Cyclophosphamide-Induced Cardiotoxicity by Inhibiting Free Radicals Generation, Inflammatory Cytokines, and Apoptotic Pathway in Rats"

_life, 2023, doi:10.3390/life13030786_

Round 1

Reviewer 1 Report

Comments for Authors

The study “Cyclophosphamide-Induced Cardiotoxicity by Inhibiting Free Radicals Generation, Inflammatory 4 Cytokines, and Apoptotic Pathway in Rats” carried out by Ahmed and Alam et al. This study highlights the role of Capsaicin on Free Radicals Generation, Inflammatory Cytokines, and Apoptotic Pathway in Cyclophosphamide-Induced Cardiotoxicity. It adds value to the existing knowledge about the important of Capsaicin in cardiac disease. Over all the study presented well and clearly written. I suggest this article for the publication with some corrections/modifications which are as follows:

1.      Line no 25 change the “mouth” with “oral

2.      Graphical Abstract: Remove (a) and (b) section and try to merge it as one figure with minimum 300dpi pixel.

3.      Line no 55 change help to “allow”

4.      In material and methods section make one sub heading, Animal model and study design

5.      Line No. 148 treatment 2 mentioned the route of administration of CYC200.

6.      Line No. 163- Replace the following with by

7.      Line no 173- capitalize the “T” in troponin-1

8.      Line no 314 write the significant value p =/</>

9.      Recommend to split figure 7 into 2 figures

10.  Line no 330 Figure 8. Remove the bold of legend

11.  Ensure the all figures are 300dpi and uniform in size, Figure should be centralized.

Author Response

To                                                                                                                    Date:3/2/2023

The Chief Editor,

Life, MDPI

Subject: Submission of the revised manuscript titled “Capsaicin ameliorates the cyclophosphamide-induced cardiotoxicity by inhibiting free radicals generation, inflammatory cytokines, and apoptotic pathway in rats” for publication

Dear Sir,

Thanks for your valuable suggestion and comments. I have gone through each comment carefully and try to correct in the manuscript and highlighted. The response to each comment of all reviewers are given in details below:

Reviewer-1 comments

Are all the cited references relevant to the research?

Response: Yes, they are. The references were well chosen based on the keywords of our research topics, and some of cited work were conducted in our laboratory (current research). If there is any issue with references, please indicate it so we can modifiy it.

Are the results clearly presented?

Response: Yes, they are. We followed the instructions provided by (MDPI-Life) instruction to the author's file. First, we described the x-axis, then the y-axis, and the trend found in the result. In the discussion section, we indicated what these results mean.

  1. Line no 25 change the "mouth" with "oral"

Response: changed to orally given

  1. Graphical Abstract: Remove (a) and (b) section and try to merge it as one Figure with minimum 300dpi pixel.

Response: Done and attached

  1. Line no 55 change help to "allow"

Response: changed to “allow to”

  1. In material and methods section make one sub heading, Animal model and study design

Response: Done as suggested.

  1. Line No. 148 treatment 2 mentioned the route of administration of CYC200.

Response: added

  1. Line No. 163- Replace the following with by

Response: changed.

  1. Line no 173- capitalize the "T" in troponin-1

Response: changed.

  1. Line no 314, write the significant value p =/</>

Response: added

  1. Recommend to split figure 7 into 2 figures

Response: Thanks for this recommendation. We were planning to split it into two figures, but we agreed that this Figure has two markers related to inflammation, so it is better to keep them as one figure for the sake of maintaining the paper story.

  1. Line no 330 Figure 8. Remove the bold of legend

Response: It is consistent with all other figure legends "bold."

  1. Ensure the all figures are 300dpi and uniform in size, Figure should be centralized.

Response: Done and attached.

*All problems have been resolved, and the final manuscript is attached.

Yours sincerely,

Dr. Mohammad Firoz Alam

Assistant Professor (Toxicology)

Department of Pharmacology and Toxicology

College of Pharmacy, Jazan University

Postal code 45142, Jazan, Kingdom of Saudi Arabia (K.S.A)

E-mail: firozalam309@gmail.com

              malam2@jazanu.jedu.sa

Reviewer 2 Report

This manuscript deals with "Capsaicin Ameliorates the Cyclophosphamide-Induced Cardiotoxicity by Inhibiting Free Radicals Generation, Inflammatory Cytokines, and Apoptotic Pathway in Rats". This article claims that using of Capsaicin could be a suitable for Cyclophosphamide-Induced Cardiotoxicity. Therefore, I suggest a minor correction and require a detailed clarification. Correction to be addressed by the authors as follows: The abstract is not well organized, where the sentences are incomplete and no continuity is there. It would be feasible, if include the significance of the current study in the abstract. A brief description of how the authors selected information from the literature in the databases, as well as what time period they searched for, is missing.

Authors should justify and expand the information on the advantages of this study for biomedical applications. Authors should specify the main experimental conditions used on the evidences from the literature. Where they briefly describe the most important data reported in the literature in a homogeneous manner and sequence reinforcing the relevance of of this approach in targeting of the senescent cell marker.

Authors should discuss whether the use of Capsaicin represents a solid alternative to existing methods. Also please discuss about the  role of mitochondria base markers.

Please add below studies to your manuscript in discussion section using below manuscripts:

DOI: 10.1155/2021/4946711

DOI: 10.1016/j.pestbp.2020.104586

DOI: 10.1016/j.arabjc.2021.103106

Conclusions should reaffirm the fundamental contribution of this paper.

Author Response

To                                                                                                                    Date:3/2/2023

The Chief Editor,

Life, MDPI

Subject: Submission of the revised manuscript titled “Capsaicin ameliorates the cyclophosphamide-induced cardiotoxicity by inhibiting free radicals generation, inflammatory cytokines, and apoptotic pathway in rats” for publication

Dear Sir,

Thanks for your valuable suggestion and comments. I have gone through each comment carefully and try to correct in the manuscript and highlighted. The response to each comment of all reviewers are given in details below:

Reviewer-2 comments

Moderate English changes required

Response:

  • The manuscript has been reviewed and double-checked by several researchers in the field. Also, using Grammarly software, we double-checked the grammar and spelling mistakes, delivery,and coherence. Consequently, these problems have been resolved (manuscript score overall, 94%).
  • Please indicate if there is any specific issue or parts need to be modified by a native speaker.

Are the methods adequately described?

  • Two out of three reviewers of this paper have no issue with the amount of information used to describe the methods.
  • We followed the same style to explain the methods in several previous published papers in the MDPI journal.
  • We added credit to those who established the protocols by referring to their detailed and well-established protocol of each experiment and just added modifications and the condition met in our laboratory.
  • We agreed that there is no need to repeat the information.

The abstract is not well organized, where the sentences are incomplete and no continuity is there. It would be feasible, if include the significance of the current study in the abstract. A brief description of how the authors selected information from the literature in the databases, as well as what time period they searched for, is missing.

Response:

  • The abstract, I think, is well organized and has been written according to the the following outline:
  • Background of cyclophosphamide (therapeutic effects and side effects)
  • Background of capsaicin (source, therapeutic effects)
  • Aim of the study
  • Study design & material, and methods
  • Results
  • Conclusion
  • Some connection words were added to the abstract to improve the flow and the unity of the abstract.
  • All sentences were checked for the correct structure of the sentence.
  • We did not include the significance of the current study in the abstract since we were limited to 250 words; however, it has been added to the discussion section, and also we added a concise sentence indicating the importance of the finding in the abstract section.
  • Writing the manuscript particularly (abstract section), we tried to make a nice story for the reader, as shown in the outline, and we also took care of the unity and the coherence, and the story of abstract when read in isolation.
  • Regarding article selection, we tried to select the most current articles, highly cited in the field, including our previously published work.
  • Regarding searching strategy, we did not write a systematic review to include detailed and a collective information about the topics. Instead, we are giving background information that generates a strong base for the reader to dive into the topic.

Authors should justify and expand the information on the advantages of this study for biomedical applications. Authors should specify the main experimental conditions used on the evidences from the literature. Where they briefly describe the most important data reported in the literature in a homogeneous manner and sequence reinforcing the relevance of of this approach in targeting of the senescent cell marker. Authors should discuss whether the use of Capsaicin represents a solid alternative to existing methods. Also please discuss about the role of mitochondria base markers.

Response: Thank you for your valuable suggestion. I have added the some information related to your quiery in discussion.

Please add below studies to your manuscript in discussion section using below manuscripts:

DOI: 10.1155/2021/4946711

Response: Some information from this article has been added in the discussion sections and it is cited.

DOI: 10.1016/j.pestbp.2020.104586

Response: Information regarding genes involved in apoptosis (mitochondria base markers) from this article has been added in the discussion sections and it is cited. if you feel we can add specific informatiom, please indicate the section or topic you think it is better to include.

DOI: 10.1016/j.arabjc.2021.103106

Response: This paper discusses role of polyphenolic compound in mentha and its mechanism of actions as anti-inflammatory, antioxidant which all have been discussed before in the manuscript. Including other information regarding mentha and its main cellular target, we feel that we will be out of topic (Caps-mechanism-cardiac markers).

Conclusions should reaffirm the fundamental contribution of this paper.

Response: It is included in the conclusion.

*All problems have been resolved, and the final manuscript is attached.

Yours sincerely,

Dr. Mohammad Firoz Alam

Assistant Professor (Toxicology)

Department of Pharmacology and Toxicology

College of Pharmacy, Jazan University

Postal code 45142, Jazan, Kingdom of Saudi Arabia (K.S.A)

E-mail: firozalam309@gmail.com

              malam2@jazanu.jedu.sa

Reviewer 3 Report

This report appears to constitute a very interesting contribution to the field of cardio-oncology.

  1. Line 17. It is not a project but a basic science research paper.
  2. How was the dose of cyclophosphamide determined? It appears to be higher than commonly applied in humans.
  3. How do you know that the source of LDH is the heart? Many different conditions result in an increase of LDH and cyclophosphamide may induce toxicity in different organs.
  4. Line 284: furthermore typing error.
  5. I don’t see the relevance of measuring triglycerides. Under conditions of inflammation, triglycerides will increase. In general, whereas it is highly plausible that the mechanism of action of capsaicin involves reduction of oxidative stress, one should be prudent to interpret all observations in a mechanistic way. Certain alterations may represent epiphenomena.

Author Response

To                                                                                                                    Date:3/2/2023

The Chief Editor,

Life, MDPI

Subject: Submission of the revised manuscript titled “Capsaicin ameliorates the cyclophosphamide-induced cardiotoxicity by inhibiting free radicals generation, inflammatory cytokines, and apoptotic pathway in rats” for publication

Dear Sir,

Thanks for your valuable suggestion and comments. I have gone through each comment carefully and try to correct in the manuscript and highlighted. The response to each comment of all reviewers are given in details below:

Reviewer-3 comments

Moderate English changes required

Response:

  • The manuscript has been reviewed and double-checked by several researchers in the field. Also, using Grammarly software, we double-checked the grammar and spelling mistakes, delivery,and coherence. Consequently, these problems have been resolved (manuscript score overall, 94%).
  • Please indicate if there is any specific issue or parts need to be modified by a native speaker.
  1. Line 17. It is not a project but a basic science research paper.

Response:  Changed

  1. How was the dose of cyclophosphamide determined? It appears to be higher than commonly applied in humans.

Response: Yes, you are right. Based on my previous study dose was selected and cited inside the text (Reference no. 28). The total dose of an individual course of cyclophosphamide therapy is a well-recognized risk factor for cardiac toxicity, but there is no consensus regarding a threshold dose. Early studies by Santos et al demonstrated considerable cardiac toxicity at doses more than 270 mg/kg over 1 to 4 days. The acute heart failure occurs in patients receiving a dose of 150 mg/kg and more (doi: 10.1177/2324709613480346). However, in many research articles, (200) (150) mg/kg on day 1 induces several types of toxicity, including cardiotoxicity, nephrotoixicty, respictivly (doi: 10.1590/s0102-865020180060000004) (https://doi.org/10.3390/ijms231911615). So in this paper we decided to go with 200 mg/kg.

  1. How do you know that the source of LDH is the heart? Many different conditions result in an increase of LDH and cyclophosphamide may induce toxicity in different organs.

Response: Yes I am agree with you and it is hard to know that cardiomyocytes is producing LDH or not as indicaction of death of cardiac cells (LDH is not cardioselective). But we also agree that LDH is very good marker for all cell or tissue damaging in the body that help to confirm the toxicity. Beside that we have use selective marker for cardiotoxicity like troponin.

  1. Line 284: furthermore typing error.

Response: Fixed.

  1. I don't see the relevance of measuring triglycerides. Under conditions of inflammation, triglycerides will increase. In general, whereas it is highly plausible that the mechanism of action of capsaicin involves reduction of oxidative stress, one should be prudent to interpret all observations in a mechanistic way. Certain alterations may represent epiphenomena.

Response:

Thanks for your suggestion and I am disagree with that triglycerides is not relvance measuring because triglycerides are an important and independent predictor of CHD and stroke risk in the Asia-Pacific region. We targeted the oxidative stress and inflammatory pathway than signaling pathway. We showed the potential mechanism of Capas by modulating several biological, pathological processes, and cardiac biomarkers. Furthermore, the cellular and biochemical changes associated with the cardiotoxicity of the CYC are attributed to the disturbance of lipid peroxidation and lipid profile (MDA & TG) level. Masuring the level of TG proives an clear picture that lipid profile is not under control as disturbance of lipid peroxidation is shown by measuring the level of MDA.

*All problems have been resolved, and the final manuscript is attached.

Yours sincerely,

Dr. Mohammad Firoz Alam

Assistant Professor (Toxicology)

Department of Pharmacology and Toxicology

College of Pharmacy, Jazan University

Postal code 45142, Jazan, Kingdom of Saudi Arabia (K.S.A)

E-mail: firozalam309@gmail.com

              malam2@jazanu.jedu.sa

Round 2

Reviewer 1 Report

The author's addressed all the queries raised in the comments.